# Consistency-based Semi-supervised Learning for Object Detection

**Jisoo Jeong**[*], **Seungeui Lee**[*], **Jeesoo Kim & Nojun Kwak**
Department of Transdisciplinary Studies
Graduate School of Convergence Science and Technology
Seoul National University
Seoul, Korea
{soo3553, seungeui.lee, kimjiss0305, nojunk}@snu.ac.kr

## Abstract

Making a precise annotation in a large dataset is crucial to the performance of object detection. While the object detection task requires a huge number of annotated samples to guarantee its performance, placing bounding boxes for every object in each sample is time-consuming and costs a lot. To alleviate this problem, we propose a Consistency-based Semi-supervised learning method for object Detection (CSD), which is a way of using consistency constraints as a tool for enhancing detection performance by making full use of available unlabeled data. Specifically, the consistency constraint is applied not only for object classification but also for the localization. We also proposed Background Elimination (BE) to avoid the negative effect of the predominant backgrounds on the detection performance. We have evaluated the proposed CSD both in single-stage[2] and two-stage detectors[3] and the results show the effectiveness of our method.

## 1 Introduction

Large datasets with complete annotations are essential to the success of object detection. Training an object detection algorithm requires annotations in the level of bounding box as shown in Fig. 1 (a). Labeling for object detection requires a pair of a category and a bounding box location for each object within each image and it is known that it takes about 10 seconds for labeling an object [1, 2]. As such, labeling for object detection consumes enormous cost, time, and effort. For example, the Caltech pedestrian detection benchmark took about 400 hours to annotate 250k images [3].

To reduce the cost of such labeling, weakly supervised learning and Semi-Supervised Learning (SSL) methods have been studied. Weakly supervised object detection methods [4, 5, 6, 7] are required to learn only with image-level labeled data, as shown in Figure 1 (b). Although this takes less efforts than the existing box-level labeling method, it results in a far inferior localization performance compared to fully supervised learning. The weakly semi-supervised object detection method [8, 9] uses fully labeled data as well as weakly labeled data, as shown in Figure 1 (c). The complete semi-supervised object detection method is to improve performance by using unlabeled data in combination with the box-level labeled data, as shown in Figure 1 (d). Studies on complete semi-supervised object detection have recently studied [10, 11] and we also deal with this problem.

Recently, there have been studies that have improved the performance of semi-supervised learning using self-training [10, 11]. They have improved the performance by utilizing high-confident samples

---

[*]Equal contribution

[2]https://github.com/soo89/CSD-SSD
[3]https://github.com/soo89/CSD-RFCN

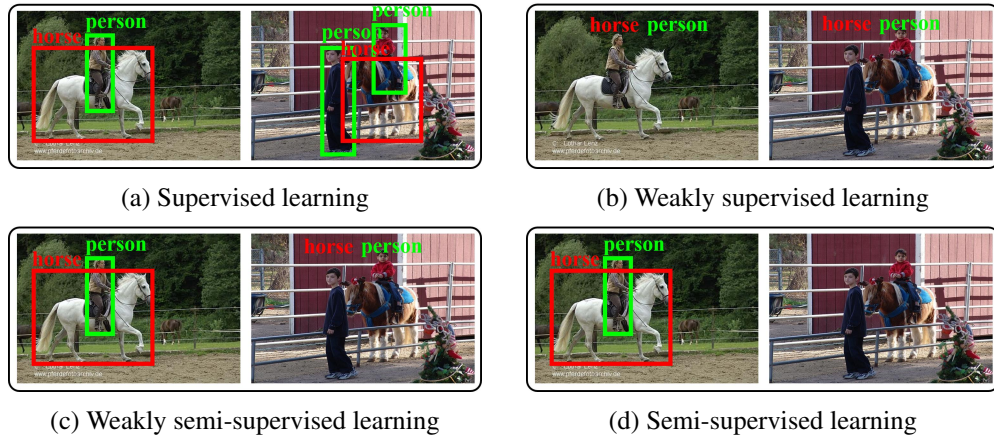

(a) Supervised learning          (b) Weakly supervised learning

(c) Weakly semi-supervised learning       (d) Semi-supervised learning

Figure 1: Different types of object detection settings

with pseudo-labels in the training. However, these methods take a long time to train because they predict all unlabeled data in each iteration and learn the gradually increasing number of training samples [12]. In addition, performances vary a lot depending on the number of pseudo-labeled samples added.

In this paper, we propose Consistency-based Semi-supervised learning for object Detection (CSD) which is similar to the consistency regularization (CR) [13, 14, 15] that has shown state-of-the-art performance in semi-supervised classification [16]. CR helps train a model to be robust to given perturbed inputs. However, it is difficult to apply CR directly to the object detection problem where multiple candidate boxes are generated for each image. Because images with different perturbation may have different numbers of boxes with various locations and sizes, it is difficult to match boxes in given images. Therefore, we use the horizontally flipped image so that one-to-one correspondence between the predicted boxes in the original and the flipped images can be easily identified. In our method, in addition to applying consistency constraint to the classification results for each predicted box, we propose a new consistency loss for fine-tuning the location of the predicted box. Experimental results show that each of these consistency losses can improve performance and we can get additional performance improvement by combining these two.

We also observed that eliminating 'background' class benefits the proposed CSD, because the predominant 'background' class affects the consistency loss much. As a way of reducing the influence of the background and achieving improved performance, we propose the Background Elimination (BE) method which excludes boxes with high background probability in the computation of consistency loss.

CSD can be applied to both the single-stage detector such as SSD [17] and the two-stage detector such as R-FCN [18]. Various ablation studies have been performed showing the benefits of the proposed consistency losses for classification and localization. Also the effect of BE has been experimentally confirmed. Experimental results show that the proposed CSD improves the detection performance for all the detectors experimented.

Our main contributions can be summarized as follows:

• We propose a novel consistency-based semi-supervised learning algorithm for object detection that can be applied not only to single-stage detectors but also to two-stage detectors.

• The proposed consistency constraints for object detection work well for both the classification of a bounding box and the regression of its location.

• We propose the BE method to mitigate the effect of background and show improvement of performance in most cases.

## 2 Related Work

### 2.1 Semi-Supervised Learning

An actual training environment usually provides a finite number of labeled data ($\mathcal{L} = \{(x_l, y_l)\}$) and an unlimited number of unlabeled data ($\mathcal{U} = \{(x_u)\}$). Many researches have tried to exploit the potential of unlabeled data since the majority of the real-world samples lack annotations. Generally, there have been two methodologies to cope with these circumstances.

**Self-training:** Self-training methods train a model using labeled data and then make predictions on unlabeled data. If the top-1 prediction score for the input $x_u$ is greater than a threshold $\sigma$, the pseudo label of $x_u$ is set as the class $\bar{y}$ whose score is the maximum. Then $x_u$ can be treated as a labeled data in the form of $(x_u, \bar{y})$ [19]. Repetitively applying this process can boost the model's performance but impedes the whole training speed. In addition, depending on the threshold value $\sigma$, the amount of added data varies a lot and this makes the performance unstable. A small number of additional pseudo-labeled samples may not improve the performance enough while too many samples may harm the performance with incorrect labeling.

**Consistency regularization:** Consistency regularization applies perturbations to an input image $x$ to obtain $x'$ and minimizes the difference between the outputs predictions $f(x)$ and $f(x')$ [13, 14, 15]. It does not require a label because the loss is determined by the difference between the outputs which is known to help smooth the manifold [16]. As mentioned above, this shows the state-of-the-art performance in semi-supervised classification problems.

### 2.2 Semi-Supervised Learning for Object Detection

**Object detection:** Object detection algorithms can be divided into two categories depending on whether a region proposal network (RPN) is used or not. Algorithms that do not use RPN are categorized as single-stage detectors, and the other algorithms are categorized as two-stage detectors. Single-stage detectors perform classification and localization in all the spatial location of feature maps [17, 20]. There have been tremendous performance improvements using deep learning and there are algorithms that are able to detect objects real-time on a desktop. Two-stage detectors are RPN-based algorithms, which detect object only for RoIs that have high possibility of containing an object [18, 21].

**Semi-supervised learning for object detection:** Until recently, most semi-supervised learning methods for object detection are based on the self-training scheme. A representative method is the Self-supervised Sample Mining (SSM) [10] algorithm, which improved performance by stitching high-confidence patches from unlabeled data to labeled data. SSM has used a method called 'evaluating consistency' to make the pseudo box label robust. It operates as a mask to verify that the Intersection over Union (IoU) score between the previously detected box and the currently detected box is greater than threshold $\gamma$. Therefore, SSM differs from our method of directly using consistency losses. SSM repeats the process of making an intermediate unlabeled data prediction and changing the training set. Consequently, SSM shares the same drawback as self-training.

## 3 Method

The CSD to be presented works differently depending whether it is for a single-stage or for a two-stage object detector. The overall CSD structure for single-stage and two-stage object detectors is depicted in Fig. 2 (a) and (b) respectively. The proposed structure is the combination of the $\Pi$-model in SSL [13] and an object detection algorithm. To allow one-to-one correspondence of target objects, an original image, $I$, and its flipped version, $\hat{I}$, are used as inputs. As in Fig. 2, a paired bounding box should represent the same class and their localization information must remain consistent.

During the training process, each mini-batch includes both labeled and unlabeled images. The labeled samples are trained using the typical object detection approach. The consistency loss is additionally applied to both the labeled and unlabeled images. In section 3.1, we explain the association method of corresponding boxes as well as the objective function used for training the object classifier in a single stage object detector. Likewise, in section 3.2, we define the objective function used for localization

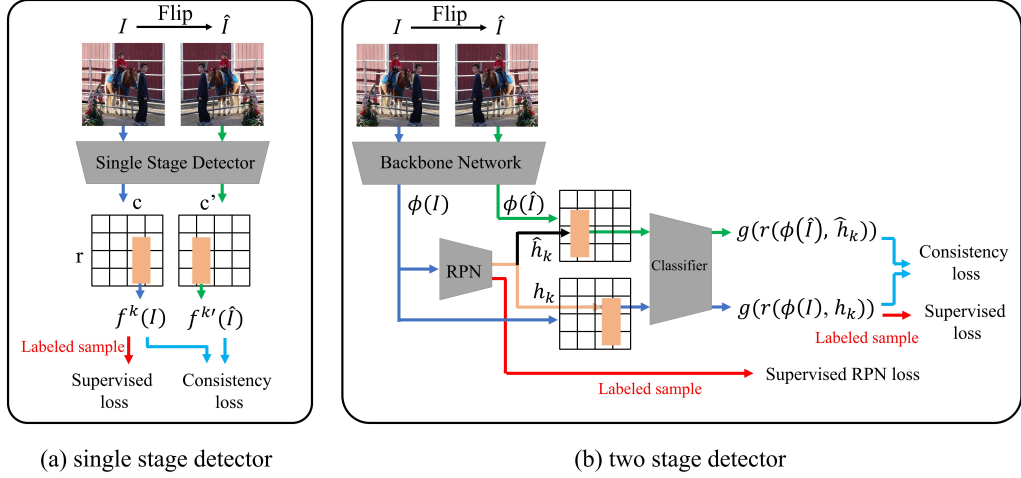

(a) single stage detector　　　　　　　(b) two stage detector

Figure 2: Overall structure of our proposed method. (a) $f^k(I)$ and $f^{k'}(\hat{I})$ are extracted by a single stage detector from image $I$ and flipped image $\hat{I}$ respectively. The supervised loss is computed between $f^k(I)$ and the ground truth for labeled data and the consistency loss is computed between $f^k(I)$ and $f^{k'}(\hat{I})$ for labeled and unlabeled data. (b) $\phi(I)$ and $\phi(\hat{I})$ originate from the backbone network and the RoI is computed only from $\phi(I)$. $\hat{h}_k$ is obtained by flipping $h_k$ to associate two corresponding boxes and supervised and consistency losses are calculated in the same way as for the single stage detector.

in both images. In the following sections afterward, we explain how these loss functions are utilized and show that our method is also applicable to a two-stage object detector.

## 3.1　Consistency loss for classification

We denote $f_{cls}^{p,r,c,d}(I)$ as the output class probability vector after softmax operation corresponding to the $p$-th pyramid, $r$-th row, $c$-th column and $d$-th default box. Since $\hat{I}$ is a horizontally flipped version of $I$, predictions of two images should be equivalent. Also we want to make these vectors, $f_{cls}^{p,r,c,d}(I)$ and $f_{cls}^{p,r,c',d}(\hat{I})$, share a very similar distribution where $c' = C - c + 1$ and $C$ is horizontal spatial dimension of the feature map. In semi-supervised learning, some candidates such as $L_2$ distance or Jensen-Shannon divergence (JSD) can be used as the consistency regularization loss. Among them, we specifically take advantage of JSD for the following reasons. $L_2$ loss treats all the classes equal and in our case, consistency loss for irrelevant classes with low probability can affect the classification performance much. We experimentally observed that the performance of SSL with $L_2$ consistency loss is even worse than that of the supervised learning. To simplify the notation, we denote the location $(p, r, c, d)$ as $k$ and the horizontally opposite location $(p, r, c', d)$ as $k'$. The classification consistency loss used for a pair of bounding boxes in our method is given as below:

$$l_{con\_cls}(f_{cls}^k(I), f_{cls}^{k'}(\hat{I})) = JS(f_{cls}^k(I), f_{cls}^{k'}(\hat{I})) \tag{1}$$

where $JS$ represents the Jensen-Shannon Divergence. The overall consistency loss for classification is then obtained from the average of loss values from all bounding box pairs:

$$\mathcal{L}_{con-c} = \mathbb{E}_k[l_{con\_cls}(f_{cls}^k(I), f_{cls}^{k'}(\hat{I}))] \tag{2}$$

## 3.2　Consistency loss for localization

The localization result for the $k$-th candidate box $f_{loc}^k(I)$ consists of $[\Delta cx, \Delta cy, \Delta w, \Delta h]$, which represent the displacement of the center and scale coefficients of a candidate box, respectively. Unlike the pair $(f_{cls}^k(I), f_{cls}^{k'}(\hat{I}))$, $f_{loc}^k(I)$ and $f_{loc}^{k'}(\hat{I})$ require a simple modification to be equivalent to each

other. Since the flipping transformation makes $\Delta \hat{c}x$ move in the opposite direction, a negation should be applied to correct it.

$$\Delta\, cx^k \Longleftrightarrow -\Delta\, \hat{c}x^{k'}$$
$$\Delta\, cy^k, \Delta\, w^k, \Delta\, h^k \Longleftrightarrow \Delta\, \hat{c}y^{k'}, \Delta\, \hat{w}^{k'}, \Delta\, \hat{h}^{k'}$$

The localization consistency loss used for a single pair of bounding boxes in our method is given as below:

$$l_{con\_loc}(f_{loc}^k(I), f_{loc}^{k'}(\hat{I})) = \frac{1}{4}(\|\Delta cx^k - (-\Delta \hat{c}x^{k'})\|^2 + \|\Delta cy^k - \Delta \hat{c}y^{k'}\|^2 \\ + \|\Delta w^k - \Delta \hat{w}^{k'}\|^2 + \|\Delta h^k - \Delta \hat{h}^{k'}\|^2) \tag{3}$$

The localization loss of each pair of bounding boxes and the total consistency loss are computed in the same principle as in the previous section:

$$\mathcal{L}_{con-l} = \mathbb{E}_k[l_{con\_loc}(f_{loc}^k(I), f_{loc}^{k'}(\hat{I}))] \tag{4}$$

### 3.3 Overall loss for object detection

The total consistency loss is composed of the losses from section 3.1 and 3.2 as in

$$\mathcal{L}_{con} = \mathcal{L}_{con-c} + \mathcal{L}_{con-l} \tag{5}$$

Eventually, the final loss $\mathcal{L}$ is composed of the original object detector's classification loss $\mathcal{L}_c$ and localization loss $\mathcal{L}_l$, in addition to the consistency loss mentioned above. As in the typical semi-supervised learning methods [13, 14], ramp-up and ramp-down techniques, which can be defined by the weight scheduling $w(t)$, are used for the stable training.

$$\mathcal{L} = \mathcal{L}_c + \mathcal{L}_l + w(t) \cdot \mathcal{L}_{con} \tag{6}$$

### 3.4 Application to two-stage detector

Unlike the single-stage detector, the two-stage detector has region proposal network (RPN) to generate region proposals and recognize the objectness of them. If we pass both the original and the flipped images to the RPN, the correspondence matching problem between the region proposals occurs which is relatively hard to solve. To simplify the problem, we only pass the feature $\phi(I)$ generated from the original image to the RPN. Then the output RoI locations are reversed and applied to the corresponding feature $\phi(\hat{I})$ as shown in Fig. 2 (b). Given the feature map $\phi(\hat{I})$ from the backbone network and the $k$-th RoI, $h_k$, from the RPN, the RoI-specific feature map of $\hat{I}$ corresponding to the flipped area $\hat{h}_k$ can be easily derived. As shown in Fig. 2 (b) RPN is trained without the consistency loss. The features corresponding to the RoI, $r(\phi(I), h_k)$ and $r(\phi(\hat{I}), \hat{h}_k)$, are processed by a classifier $g$. Then, outputs $g(r(\phi(I), h_k))$ and $g(r(\phi(\hat{I}), \hat{h}_k))$ are used to compute the loss to train the network. As will be seen in the experiments, compared to the single-stage detector, the performance improvement of the proposed CSD is lower for two-stage detector and this attributes to the lack of consistency loss in RPN training.

### 3.5 Background elimination

Particularly in object detection, an additional class of 'background' exists and most of the candidate boxes are usually classified to this class unless it is filtered by a confidence threshold. Consequently, consistency losses computed with all candidates will be easily dominated by backgrounds. This can degrade the classification performance for the foreground classes. Therefore, we exclude boxes having a high probability of background class by marking it with a mask. The mask is created according to the classification result for every candidate bounding box of $I$ as in

$$m^k = \begin{cases} 1, & \text{if } \text{argmax}(f_{cls}^k(I)) \neq background \\ 0, & \text{otherwise.} \end{cases} \tag{7}$$

Applying the mask to (2) and (4) yields

$$\mathcal{L}_{con-c} = \mathbb{E}_{\mathbb{I}_{m^k=1}}[l_{con\_cls}(f_{cls}^k(I), f_{cls}^{k'}(\hat{I}))], \quad \mathcal{L}_{con-l} = \mathbb{E}_{\mathbb{I}_{m^k=1}}[l_{con\_loc}(f_{loc}^k(I), f_{loc}^{k'}(\hat{I}))] \tag{8}$$

where $\mathbb{I}_{m^k=1}$ indicates that the expectation is taken only for the positive mask.

Table 1: Detection results for PASCAL VOC2007 test set. The first two rows show the performance of each detector by supervised learning. * is the score from [17, 18]. The following experiments use VOC07 as the labeled data and VOC12 as the unlabeled data, and show the results of the proposed CSD with/without $\mathcal{L}_{con-c}$ (cls), $\mathcal{L}_{con-l}$ (loc) and EB. Blue / Red : supervised score (baseline) and Best results. The numbers in the parentheses are the performance enhancement over the baseline.

| Labeled data | Unlabeled data | Consistency | | Background Elimination | mAP (%) | | |
|---|---|---|---|---|---|---|---|
| | | cls | loc | | SSD 300 | SSD 512 | R-FCN |
| VOC07 | - | - | - | - | 68.0*/70.2 | 71.6*/73.3 | 73.9 |
| VOC0712 | - | - | - | - | 74.3*/77.2 | 76.8*/79.6 | 79.5*/79.4 |
| VOC07 | VOC12 | ✓ | - | - | 71.6 (1.4) | 74.6 (1.3) | 74.0 (0.1) |
| | | - | ✓ | - | 72.2 (2.0) | 74.6 (1.3) | 73.9 (0.0) |
| | | ✓ | ✓ | - | 72.0 (1.8) | 74.8 (1.5) | 74.0 (0.1) |
| VOC07 | VOC12 | ✓ | - | ✓ | 71.7 (1.5) | 75.4 (2.1) | 74.5 (0.6) |
| | | - | ✓ | ✓ | 71.9 (1.7) | 75.2 (1.9) | 74.4 (0.5) |
| | | ✓ | ✓ | ✓ | 72.3 (2.1) | 75.8 (2.5) | 74.7 (0.8) |

## 4 Experiments

In our experiments, we have utilized the PASCAL VOC [22] and MSCOCO [23] datasets which are the most popular datasets in object detection. They consist of 20 and 80 classes respectively. PASCAL VOC 2007 and 2012 datasets consist of 5,011 and 11,540 trainval (train and validation) images respectively. In this paper, PASCAL VOC2007 trainval is used as the labeled data and PASCAL VOC2012 trainval and MSCOCO are utilized as the unlabeled one. We use test set of PASCAL VOC2007 (4,952 images) for testing.

The codes used for our experiments are based on Pytorch. We have used third-party codes for SSD [17] [4] and R-FCN [18] [5]. All experiments have been done under the similar setting with the code[6] of the author. Expediently, labeled and unlabeled data are gathered in a single dataset and then randomly shuffled. In our setting, both labeled and unlabeled samples sit together in each mini-batch. The experimental settings of R-FCN are referred to those of SSM. As the batch size used for R-FCN is $4$, using the same sampling strategy of SSD experiments does not guarantee that at least one labeled data is included in every mini-batch. To solve this problem, we have established separate data-loaders for labeled and unlabeled data. The amount of unlabeled data in a mini-batch is three times larger than that of the labeled data[7]. The total number of RoIs for CSD is 2k and all the parameter settings and training details are presented in the supplementary material.

### 4.1 Ablation Study

We have examined the influence of $\mathcal{L}_{con-c}$, $\mathcal{L}_{con-l}$ and Background Elimination (BE) on SSD300, SSD512 and R-FCN and the performances are presented in Table 1. For SSD300, supervised learning using VOC07 and VOC0712 show 70.2 mAP and 77.2 mAP respectively. Using $\mathcal{L}_{con-c}$ with Jensen-Shannon divergence induces 1.4% of improvement while $\mathcal{L}_{con-c}$ with $L_2$-norm causes a performance degradation to 70.0 mAP, which is slightly lower than that of the supervised learning. $\mathcal{L}_{con-l}$ shows 2.0% of enhancement and jointly using both consistency losses shows 1.8% of enhancement. Particularly in SSD300 using $\mathcal{L}_{con-l}$ only has shown better performance than using both. SSD512 scored 73.3 mAP and 79.6 mAP in pure supervised learning on VOC07 and VOC0712 respectively. Separate use of $\mathcal{L}_{con-c}$ or $\mathcal{L}_{con-l}$ induces 1.3% of improvements in both cases and joint usage of both losses improves 1.5% of accuracy. BE significantly improves the performance when used with both of the consistency losses. Especially, since more regions are predicted as backgrounds in SSD512 compared to SSD300, BE is more beneficial to SSD512 than to SSD300.

As mentioned in section 3.4, CSD in R-FCN uses consistency losses only after the RoI pooling and not in the RPN. For R-FCN, supervised learning using VOC07 and VOC0712 shows 73.9 mAP and 79.4 mAP of accuracy respectively. There are small or no performance improvement before

Table 2: Detection results on PASCAL VOC2007 test set. "COCO$^\S$": All 80 classes. "COCO$^\dagger$": 20 PASCAL VOC classes.

| Labeled data | Unlabeled data | CSD Method (mAP) | | |
|---|---|---|---|---|
| | | SSD300 | SSD512 | R-FCN |
| VOC07 | - | 70.2 | 73.3 | 73.9 |
| VOC07 | VOC12 | 72.3 | 75.8 | 74.7 |
| | VOC12+COCO$^\S$ | 71.7 | 75.1 | 74.9 |
| | VOC12+COCO$^\dagger$ | 72.6 | 75.9 | 75.1 |

Table 3: Effects of using Background Elimination (BE) on SSD300 performance.

| VOC07(L)+VOC12(U) | mAP |
|---|---|
| without BE | 72.0 |
| BE with $m^k$ | 72.3 |
| BE with $m^k \otimes m^{k'}$ | 71.7 |

applying BE. However, when BE is applied, performance is improved by adding $\mathcal{L}_{con-c}$ and $\mathcal{L}_{con-l}$. In addition, the performance is further improved with simultaneous use of both consistency losses.

## 4.2 Unlabeled data with different distribution (MSCOCO)

To see the effect of unlabeled data with different distribution to the labeled set, we use VOC07 as the labeled data and VOC12 plus MSCOCO as the unlabeled data as shown in Table 2. We denote 'trainval' of the MSCOCO dataset (123,287 images) as COCO$^\S$ and the dataset (19,592 images) of which images contain only objects belonging to the 20 PASCAL VOC classes as COCO$^\dagger$. Details on learning scheduling are in the supplementary material.

In single-stage detectors, the performance by training with unlabeled VOC12 and COCO$^\S$ shows better performance than the supervised learning, but it is less than the performance using unlabeled VOC12 only. In a two stage detector, it shows higher performance than the supervised learning and training with unlabeled VOC12 data. Training with unlabeled VOC12 and COCO$^\dagger$, both the single-stage detector and two-stage detector show performance improvements. We analyze this phenomenon in the next section.

## 5  Discussion

**Consistency regularization with only labeled data:** We evaluated our method on PASCAL VOC 2007 under the supervised training setting. We observed that training with the consistency loss only on labeled data led to worse results. It means that the consistency loss does not affect the improvement of performance for labeled data. Our consistency loss has a regularization effect on the CNN filters such that they are enforced to be symmetric. We conjecture this reduces the representation power of CNNs and causes performance degradation when used with ground truth labels. However, as shown in Table 1, our consistency constraints are helpful to improve the performance for semi-supervised object detection task. The results of these experiments are provided in the Supplementary Material.

**Single-stage detector vs. Two-stage detector:**  We apply consistency constraint differently depending on whether RPN is used or not. First, in a single-stage detector, the proposed consistency losses can be applied to all areas and it shows much improvement in performance. The two-stage detector, on the other hand, uses $\hat{h}$ by flipping the $h$ obtained from $I$. Therefore, while we can expect to improve performance in the classifier, it is hard to expect additional performance improvement of RPN. As a result, the two-stage detector has less performance improvement than the single-stage detector. To optimize the RPN, a new way exploiting the consistency loss is needed, which we leave as further work.

**Background Elimination:** The proportion of background in the predefined boxes is very large. We apply BE to reduce the effect of the background and show that BE is helpful in improving the performance. However, getting rid of too many samples is not helpful in learning, as shown in Table 3. As a way of reducing more background samples, the consistency losses are applied to the candidate boxes only when their estimated class is non-background ($m^k = 1$) as well as their flipped boxes on the flipped images are estimated as non-background ($m^{k'} = 1$). At this time, the performance of the SSD300 model shows 71.7 mAP, which is 0.6% lower than the original 72.3 mAP. This shows that removing too many background samples may cause performance degradation.

**Datasets:** Table 2 shows that in learning 20 classes of VOC, additional usage of unlabeled data leads to an enhanced performance. However, the ratio of labeled/unlabeled class mismatch decides the

Table 4: Comparisons between self-training and consistency regularization based methods on PASCAL VOC2007 test set. "COCO$^\S$": All 80 classes. "COCO$^\dagger$": 20 PASCAL VOC classes.

| Single-Stage Detector | | | | |
|---|---|---|---|---|
| Method | Labeled data | Unlabeled data | mAP | Gain |
| SSD512 (supervised) | VOC07 | - | 73.3 | - |
| SSD512 + CSD (ours) | VOC07 | VOC12 | 75.8 | 2.5 |
| SSD512 + CSD (ours) | VOC07 | VOC12 + COCO$^\S$ | 75.1 | 1.8 |
| SSD512 + CSD (ours) | VOC07 | VOC12 + COCO$^\dagger$ | 75.9 | 2.6 |
| Two-Stage Detector | | | | |
| Method | Labeled data | Unlabeled data | mAP | Gain |
| R-FCN (supervised) | VOC07 | - | 73.9 | - |
| RFCN + SPL (300%) [24] | VOC07 | | 74.1 | 0.2 |
| RFCN + SPL (400%) [24] | VOC07 | VOC12 + COCO$^\S$ | 74.7 | 0.8 |
| RFCN + SSM (300%) [10] | VOC07 | | 75.6 | 1.7 |
| RFCN + SSM (400%) [10] | VOC07 | | 76.7 | 2.8 |
| RFCN + CSD (ours) | VOC07 | VOC12 | 74.7 | 0.8 |
| RFCN + CSD (ours) | VOC07 | VOC12 + COCO$^\S$ | 74.9 | 1.0 |
| RFCN + CSD (ours) | VOC07 | VOC12 + COCO$^\dagger$ | 75.1 | 1.2 |

amount of improvement. This is why the case of using VOC12 + COCO$^\dagger$ shows a better result than the case of VOC12 + COCO$^\S$. This result is consistent with the recent study by [16].

BE is hardly expected to remove this out-of-distribution. It is intended to eliminate background, but classes in MSCOCO can have a higher confidence in other classes that are similar. For example, classes such as 'giraffe' and 'elephant' may have features similar to 'horse' or 'dog' rather than the background. These data can interfere with training detectors.

On the other hand, adding unlabeled data with a similar distribution, all detectors have improved the performance. Our CSD does not need any labeling in the additional data but it still has its limitation that the distribution of the unlabeled data should be similar to that of the labeled data. Further research is needed to solve this problem, which we leave it for future work.

**Self-training vs. Consistency regularization:** Self-training is widely used as a simple heuristic method in semi-supervised learning. As it is an iterative method which cycles training, prediction of unlabeled data and changing the training dataset, it is time-consuming and computationally intensive [12]. In addition, the threshold and stop criterion, which decide the quality and quantity of an additional dataset, affect the algorithm's performance. Meanwhile, CR method which trains unlabeled data with an additional loss helps the more common and robust learning.

Table 4 shows the performance of SPL [24], SSM [10] and CSD. SPL and SSM are based on the self-training method, which shows different performance depending on the amount of data added [8]. They experimented only with R-FCN framework and trained with VOC07 as labeled data and VOC12 and MSCOCO as unlabeled data. The performance of SPL is improved by 0.2 and 0.8 than baseline while SSM has 1.7 and 2.8 better performance than the baseline. In CSD, according to unlabeled dataset, it shows performance improvement of $0.8 \sim 1.2$ than baseline. As mentioned above, CSD has a limitation in the two stage detector, which has less performance improvement than single stage detector. In single stage detector, however, SSD512 shows the 1.8% and 2.6% performance improvements.

## 6   Conclusion

In this paper, we have introduced a novel Consistency-based Semi-supervised learning for object Detection (CSD) method. To the best of our knowledge, it is the first attempt to extend CR used in conventional semi-supervised classification problems to object detection problem. We applied the proposed CSD to single-stage detectors and a two-stage detector respectively and designed loss to improve the performance of both detectors over the supervised learning method. We have shown that consistency loss is helpful for semi-supervised learning in classification as well as localization with various ablation experiments. In addition, BE has been shown to improve performance.

# 7 Acknowledgments

This work was supported by IITP grant funded by the Korea government (MSIT) (No.2019-0-01367) and Next-Generation Information Computing Development Program through the NRF of Korea (2017M3C4A7077582).

## Footnotes

[4]https://github.com/amdegroot/ssd.pytorch

[5]https://github.com/princewang1994/R-FCN.pytorch

[6]https://github.com/weiliu89/caffe/tree/ssd

[7]During the training, we allow the labeled data and unlabeled data not to share the same epoch number.

[8] % means that the percentage of additional unlabeled objects over labeled objects.

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
