[Supplementary Material]

# Consistency-based Semi-supervised Learning for Object Detection
# Supplementary Materials

## 1 S1 Implementation Detail

**2 Single-stage detector :**

3 Both SSD300 and SSD512 are used in our experiments. The authors of SSD provide the code in a
4 github repository and we have followed the experimental settings in it. The backbone network is
5 VGG16 pre-trained with ImageNet dataset. With PASCAL VOC dataset, models in all experiments
6 have been trained for 120k iterations with a mini-batch size of 32. The learning rate is multiplied by
7 0.1 at 80k and 100k iterations. For the weight scheduling function $w(t)$, we have followed the policy
8 of temporal ensembling [1]. The function is defined as below:

$$w(t) = \begin{cases} exp^{-5 \times (1-\frac{t}{t_1})^2}, & t < t_1 \\ 1, & t_1 \le t < T - t_2 \\ exp^{-12 \times (1-\frac{T-t}{t_2})^2}, & t \ge T - t_2. \end{cases} \quad (1)$$

9 Here, $T$ denotes the total number of iterations while $t_1$ and $t_2$ represent the ramp-up and ramp-down
10 coefficients respectively. In experiments using PASCAL VOC dataset, $t_1$ is set to 32k and $t_2$ is set to
11 20k. Experiments of VOC-only COCO (COCO$^\dagger$) and Full COCO(COCO$^\S$) have been done with 240k
12 and 360k of iterations, and parameter scheduling is also changed $2 \sim 3$ times, excluding ramp-down.
13 If the ramp-down is longer, the influence of consistency is less affected by learning. Therefore, the
14 ramp-dwon is maintained. A proper organization of a single mini-batch plays an important role for
15 stable training. In case of COCO$^\S$, the ratio of labeled data to unlabeled data is 1:26 which means
16 that labeled data may be omitted in a mini-batch. To compensate this, unlabeled data are sampled
17 according to the total number of samples in VOC12.

18 **Two-stage detector :** In experiments using a two-stage detector, we have adopted R-FCN under the
19 same setting of SSM. ResNet-101 is used as a backbone network. In experiments using PASCAL
20 VOC dataset, every model has been trained for 70k iterations with a batch size of 4. The learning rate
21 is multiplied by 0.1 at 50k iterations. $t_1$ is set to 20k and $t_2$ is set to 10k. Experiments of VOC-only
22 COCO (COCO$^\dagger$) and Full COCO(COCO$^\S$) have been done with 140k and 210k of iterations, and
23 parameter scheduling is also changed $2 \sim 3$ times, excluding ramp-down. In R-FCN, the ratio of
24 positive regions to negative regions is 1:3 and this rate is applied to COCO as well.

## 25 S2 Effectiveness of consistency regularization for labeled dataset

26 To verify whether the performance improvement comes from Consistency regularization(CR) or the
27 inflow of unlabeled data, we present experimental results applying CR on the supervised learning
28 setting. As a result, all models scored worse than the supervised one without CR. Therefore, CR

Table S1: Detection results for PASCAL VOC2007 test set. The first two rows show the performance of each detector by supervised learning. * is the score from [2]. The following experiments use VOC07 as the labeled data and show the results of the proposed CSD with/without $\mathcal{L}_{con-c}$ (cls), $\mathcal{L}_{con-l}$ (loc) and EB. Blue : supervised score (baseline)

| Labeled data | Unlabeled data | Consistency | | Eliminating Background | Method (mAP) SSD300 |
|---|---|---|---|---|---|
| | | cls | loc | | |
| VOC07 | - | - | - | - | 68.0*/70.2 |
| VOC0712 | - | - | - | - | 74.3*/77.2 |
| VOC07 | - | ✓ | - | - | 69.4 |
| | | - | ✓ | - | 69.9 |
| | | ✓ | ✓ | - | 69.7 |
| VOC07 | - | ✓ | - | ✓ | 70.2 |
| | | - | ✓ | ✓ | 69.8 |
| | | ✓ | ✓ | ✓ | 69.3 |

does not work in a supervised learning method and unlabeled data is essential for performance enhancement.