[Reviews · NeurIPS 2019]

Reviewer 1



The paper presents a semi-supervised approach for object detection that extends the consistency regularization used for image classification [14] for object detection. Concretely, it proposes using consistency losses for both classification and localization, as well as a background elimination technique that alleviates the class imbalance inherent to object detection. They evaluate their approach with two types of detectors (single and two-stage) on PASCAL VOT 2007 with unlabeled data from VOT2012 and COCO. Pros: + The approach is novel, as far as I know no previous work addresses semi-supervised learning with consistency regularization for object detection. + The use of JS divergence over L2 distance is justified and shown experimentally. However, it is not clear to me that conventional semi-supervised learning always uses L2. For example, [14] mentions KL divergence as a possible distance for consistency regularization. + Background estimation is a nice contribution to provide a better adaptation for object detection. It is well motivated and it seems to work well in practice. As a suggestion, another option would be not applying CR on very uncertain samples (e.g. high entropy). This might help when the unlabeled dataset follows a different data distribution. + Convincing results with respect to fully supervised baseline. Adequate ablation showing the effect of each contribution and some analysis of the results, although they could provide more insights on why the combination of con-c and con-l is not always beneficial. + Good discussion section and nice experiment with two COCO versions. + Overall, the paper is clearly written and fig. 2 is informative. Cons: - Technically, it is a bit incremental as this type of loss already existed for image classification and the extension to localization is rather trivial. - The related work section is not very complete in terms of number of cited related works. Some papers that come to mind: Tang et al. CVPR16 "Large Scale Semi supervised Object Detection using Visual and Semantic Knowledge Transfer", and Doersch and Zisserman ICCV17 "Multi-task self-supervised visual learning". - The method is a bit limited when it comes to two-stage detectors, as the RPN is not trained in a semi-supervised manner. The authors could consider (maybe in future work) to apply an analogous treatment for the anchor windows inside the RPN. - I find it curious that CR doesn't work on a fully supervised setting but that it is applied for labeled samples for semi-supervised. I would like to see what happens when the CR is only applied to unlabeled samples in this case. - A few grammar errors and typos. Post rebuttal: The authors addressed many of my concerns and those of other reviewers in the rebuttal. I am still a bit intrigued by the effect of CR on labeled samples but I appreciate the extra experiment. I keep my acceptance score.

Reviewer 2



Originality and significance: In my opinion the direction this paper is exploring is very important and worth investigating. It is hard to judge the significance of this paper since the comparison with other methods in the experimental evaluation is very brief and no conclusion can be inferred based on them. Quality: * I don't think that provided experimental evaluation is enough to establish the paper contribution over the prior work. For R-FCN detector SSM [8] approach demonstrates significantly better performance. It would be beneficial to see SSD + SSM comparison with the proposed approach. Given that the code for SSM is public, it should be possible to conduct such experiments. * In the paper the authors claim that using COCO images that contain only 20 Pascal VOC categories is beneficial since they have the same distribution as Pascal VOC. I do not agree with this claim. Pascal VOC certainly has objects from the other 60 categories in it. They simply do not have annotation for them. Filtering out COCO like this one uses additional supervision by supplying images that has objects from these 20 categories and do not have objects of other 60 categories. In my opinion, the paper's claim should be refined or the experiments with COCO subset should be removed. Clarity: The paper explains the new approach clearly. I think it would be possible for me to reproduce paper's results.

Reviewer 3



This paper describes an approach to semi-supervised learning for object detection in images. The authors propose two consistency losses (one for localization base on MSE, and the other for classification based on JS-divergence between predicted class distributions). This consistency loss is applied to original and horizontally flipped versions of labeled and unlabeled images during training. The hypothesis being that any predicted, localized objects should be invariant to flipping. The proposed loss can be incorporated into a variety of state-of-the-art object detection architectures and consistently improves downstream performance (with some caveats). Experimental results are given on PASCAL 2007 using unlabeled data from PASCAL 2012 and MSCOCO. The main selling point of this work is the simplicity of the proposed regularization and its applicability across architectures. However, there I have two main concerns with the work: 1. Datasets. Although PASCAL 2007 is a venerable benchmark dataset for object detection, it is definitely showing its age. Moreover, the PASCAL 2012 dataset has the same class distribution as PASCAL 2007, and MSCOCO has significant overlap. The authors clearly acknowledge this (and split MSCOCO into two distributions for experiments). However, *both* PASCAL 2012 and MSCOCO are highly curated object detection datasets, and semi-supervised learning for object detection would be far more convincing if the "unlabeled" data used were sampled from more arbitrary sources. 2. Comparison with Self-supervised Sample Mining (SSM) [8]. The comparison with SSM in Table 4 seems to indicate that it does not share "the same drawback as self-training" as stated in the related work. Since the results of the proposed consistency-based approach are very similar, a deeper analysis of the differences and advantages should be provided. Why should one prefer consistency over SSM? POST REBUTTAL: The authors touched on all of my concerns in the rebuttal. Though I still think the work would be significantly improved using less curated sources of unlabeled data, the paper is solid and the results on PASCAL 2007 convincing.

Reviewer 4



My major concern lies in the experimental configurations. The authors use the VOC07 test set for evaluation and use the VOC07 trainval (labeled) + VOC12/COCO (unlabeled) for training. As the VOC dataset is less challenging and has gradually fallen out of fashion in recent years, in my opinion, a fairer evaluation should be made on COCO dataset, say, by partially removing the annotations of the training data as auxiliary ones. Besides, I also notice that the best results of the proposed method are obtained when the auxiliary COCO images only contain objects belonging to the 20 VOC classes (see table 2 and table 4). In this case, the comparisons would not be fair enough since the authors are using manually selected data and the class labels are essentially considered during training. At least, in my opinion, I do not agree considering the above sensoria as a “semi-supervised” detection setting. The idea of alleviating the class-imbalance problem in object detection by using “background elimination (BE)” is not new. If the authors insist considering it as one of the contributions, they should better compare it with other approaches with similar motivations, e.g. hard negative mining in SSD and Focal loss in RetinaNet. The authors simply use horizontal flip as the perturbation of an input image so that to achieve a one-to-one correspondence between the predicted boxes in the flipped image and the original ones. My question is why not try other operations, like illumination changes, image wrap, and minor rotation, where such correspondence can also be easily guaranteed. Will a more diverse perturbation be beneficial to improve performance?

[Author Response · NeurIPS 2019]

Thanks for your constructive reviews. We tried our best to properly respond to all the inquiries from the reviewers.

## 1. Common Response

**Comparison with SSM:** For a practical usage of a model, the stopping criteria must be defined properly. SSM continues the training until no more unlabeled data are included in the training set (Algorithm 1 in [1]). In Fig.4 of [1], the performance improves initially as unlabeled samples are added but it starts to degrade as more samples are added. However, the reported score of the SSM is not from an objective stopping criterion but the peak performance during the entire iterations. With this setting, the influx of the data from out-of-distribution and incorrectly labeled samples cannot be prevented. As we all know that the performance of SSM should not be measured with this setting, we are not sure that the performance of a detector combined with SSM would get better. To check this, we tried to implement SSM in SSD. However, many details are missing in SSM and the learning parameters of single-stage detector and two-stage detector are different. Its intense time-cost and huge hyper-parameter space makes it difficult to implement SSM properly. In our work, we just wanted to present a representative self-learning method.

**Dataset:** When the unlabeled samples originate from out-of-class distribution, the performance of any semi-supervised learning method usually degrades. This is a well known limitation of SSL [2] and we wanted to show the performance of our method in the in-class distribution setting as other works.
Also, we acknowledge that using a significantly huge and unrefined data would be an ideal setting. However, this paper has its focus on applying the consistency-based semi-supervised approach to the object detection task and the experimental settings have been inspired from those in the paper of SSM. We would like to mention on the dataset-related research direction that some reviewers pointed out in an additional paragraph in the final version.

## 2. Response to each Reviewer

**Reviewer 1:** In consistency regularization, various candidates of loss function can be used. We simply chose the $\Pi$ model, which uses $L_2$ loss, as a baseline and we will explain this more in detail in the final version.
Particularly in the case of using con-l in SSD300, the performance is quite good even without using BE. We think that it is because the number of samples with flat areas is relatively small for a small input resolution and this helps the regression process while con-c disturbs the effect of con-l.
We have experimented with SSD300 model for the effect of not using CR on labeled samples. SSD300 (CR on unlabeled samples only) scored 72.1mAP in semi-supervised learning using VOC07(labeled) and VOC12(unlabeled) and it is slightly worse than the base CSD (72.3 mAP).
We will search for more related works including the ones the reviewer mentioned and will try to explore a link between the works and Fig. 1 in our paper.

**Reviewer 2:** Table 2 shows the results of experiments with 20 COCO classes and the entire 80 classes. As the reviewer mentioned, we would like to modify our claim (line 246-248) more clearly as follows: The reason why using 20 class categories as unlabeled data outperforms the other case is because the ratio of *labeled/unlabeled class mismatch* is much smaller. This is similar to the experimental results of Fig. 2 in [2].

**Reviewer 3:** As mentioned in the paper, the self-training method is an iterative method. So it is time-consuming and computationally intensive. In addition, the mAP score is reported wrongly in SSM, as mentioned above. Meanwhile, our CSD is a new type of semi-supervised learning method for object detection that can be trained with an additional loss reflecting consistency between pairs of images.

**Reviewer 4:** Both hard negative mining and focal loss require a label in their training. To apply these to the consistency loss of unlabeled data, they should be modified appropriately. Therefore, we propose BE, which is applicable in the semi-supervised learning environment. It seems that the reviewer considered the weighted loss using the background score, which we think can be effective that can be another new contribution. We will apply it in our future research.
It is important to clarify the relationship between boxes by image perturbation to calculate the consistency loss. As the reviewer mentioned, the performance is expected to be improved by diversity if some limited perturbation schemes are applied. Although some of the perturbation methods the reviewer mentioned have minor problems (e.g image wrapping can cut off an object, which is not representable by a single bounding box), we agree that additional perturbation methods will probably improve the performance. However, flipping is the most simple method which always guarantees the one-to-one correspondence of given boxes. In the discussion, we will complement the possibility of using other perturbation methods.

## References

[1] Keze Wang, Xiaopeng Yan, Dongyu Zhang, Lei Zhang, and Liang Lin. Towards human-machine cooperation: Self-supervised sample mining for object detection. In *CVPR*, pages 1605–1613, 2018.

[2] Avital Oliver, Augustus Odena, Colin A Raffel, Ekin Dogus Cubuk, and Ian Goodfellow. Realistic evaluation of deep semi-supervised learning algorithms. In *NIPS*, pages 3235–3246, 2018.


[Meta-Review · NeurIPS 2019]

This paper introduces a semi-supervised approach for object detection that extends the consistency regularization used for image classification for object detection. The proposed approach is novel and interesting. The evaluation part can be improved to make the comparison more convincing, as suggested by several reviewers.